# How *Staphylococcus aureus* and *Pseudomonas aeruginosa* Hijack the Host Immune Response in the Context of Cystic Fibrosis

**DOI:** 10.3390/ijms24076609

**Published:** 2023-04-01

**Authors:** Aubin Souche, François Vandenesch, Anne Doléans-Jordheim, Karen Moreau

**Affiliations:** 1Centre International de Recherche en Infectiologie, Inserm, U1111, Université Claude Bernard Lyon 1, CNRS, UMR5308, ENS de Lyon, 69007 Lyon, France; 2Institut des Agents Infectieux, Hospices Civils de Lyon, 69002 Lyon, France

**Keywords:** *S. aureus*, *P. aeruginosa*, immune response, cystic fibrosis

## Abstract

Cystic fibrosis (CF) is a serious genetic disease that leads to premature death, mainly due to impaired lung function. CF lungs are characterized by ongoing inflammation, impaired immune response, and chronic bacterial colonization. *Staphylococcus aureus* (SA) and *Pseudomonas aeruginosa* (PA) are the two most predominant bacterial agents of these chronic infections. Both can colonize the lungs for years by developing host adaptation strategies. In this review, we examined the mechanisms by which SA and PA adapt to the host immune response. They are able to bypass the physical integrity of airway epithelia, evade recognition, and then modulate host immune cell proliferation. They also modulate the immune response by regulating cytokine production and by counteracting the activity of neutrophils and other immune cells. Inhibition of the immune response benefits not only the species that implements them but also other species present, and we therefore discuss how these mechanisms can promote the establishment of coinfections in CF lungs.

## 1. Introduction

Cystic fibrosis (CF) is a severe autosomal recessive genetic disease that affects around 70,000 patients worldwide, mostly in the Caucasian population in which the incidence reaches 1/2500 births [1]. Despite high-intensity of medical care, the mean age of death is still only 33.9 years in the U.S. in 2021 and 42.9 years in France in 2022 [2,3]. CF is caused by mutations in the gene coding for the cystic fibrosis transmembrane conductance regulator (CFTR) ion channel located at the apical pole of epithelial cells which plays a key role in mucus homeostasis. CF is characterized by chronic inflammation of the airways due to both CFTR deficiency [4] and bacterial infection. This chronic inflammation leads to lung damage and reduced respiratory function, responsible for clinical worsening.

*Staphylococcus aureus* (SA) and *Pseudomonas aeruginosa* (PA) are the most prevalent pathogens chronically colonizing CF airways [5,6]. In young CF patients, SA are the most prevalent bacteria, and then, during early adulthood, PA becomes the most prevalent bacteria which is associated with worse clinical course [7]. This kinetic is at the origin of the dogma according to which PA replaces SA. However, this hypothesis has not been confirmed by Fischer et al. [8]. It is estimated that between 8.6% to 60% of CF patients are co-colonized by the two bacteria [6,7]. In this review, we address the question of how the two major pathogens in CF can influence the host immune response and how this can promote chronic coinfections. The effects of SA and PA on host immune cells are, respectively, summarized in Table 1 and Table 2.

## 2. Modulation of Physical Integrity of Airway Epithelium

During airway colonization and infection, the first host defense that bacteria encounter is the physical barrier constituted by the airway epithelium; in addition, airway epithelium is also responsible for mucociliary clearance. In CF patients, this mucociliary clearance is impaired due to poorly hydrated mucus, favoring bacterial adherence. PA is also able to act on this barrier by producing CFTR inhibitory factor (Cif) that induces CFTR degradation (if still present on the apical pole, which is dependent on the CF mutation [76,77]) and contributes to mucus thickening and then reinforces the impairment of mucociliary clearance [44]. PA also secrets quorum sensing (QS) molecules, such as N-3-oxododecanoyl-L-homoserine lactone (3O-C12-HSL), that induce epithelium tight junction disruption, the destruction of the adherens junction, and apoptosis in airway epithelial cells [45,78], as well as the type III secretion system (T3SS) effectors ExoS, ExoT, ExoY, and ExoU that are responsible for cytoskeleton destruction and cell retractation [78]. Additionally, ExoU is responsible for rapid eukaryotic cell death, including epithelial barriers [46]. PA also produces a protease, LasB, that contributes to the destruction of junctional proteins [78]. SA is also able to disrupt epithelial barrier via Hla production, a pore forming toxin that targets the ADAM10 receptor on epithelial cells [9]. The destruction of these cells may lead to inhibition of host immune response by reducing epithelial cell cytokine production, favoring bacterial persistence.

## 3. Modulation of Immune Cell Proliferation and Death

SA and PA are able to interfere with immune cells, both by activation of apoptosis or inhibition of proliferation. On the one hand, AdsA, a SA protein responsible for adenosine production by ATP, ADP, and AMP degradation, leads to deoxyadenosine formation, a metabolite responsible for caspase-3 apoptosis induction in macrophages [30,38]. AdsA, is also responsible for T cell activation and proliferation inhibition due to adenosine accumulation [15]. In parallel, staphylococcal protein A (SpA), a sortase-anchored surface protein of SA, which has high affinity for human immunoglobulins (IgA, IgD, IgG1-4, IgM and IgE), promotes B lymphocyte proliferation and their apoptotic collapse [36]. Furthermore, SA is able to kill host immune cells, such as neutrophils, by producing pore-forming toxins such as Hla, HlgAB, HlgCB, Panton–Valentine leucocidin, and other bicomponent leukocidins, leading to immune evasion [9,37]. On the other hand, PA 3O-C12-HSL and PQS molecules are able to inhibit the proliferation of peripheral blood mononuclear cells (PBMC) and mast cells [47,48], leading to reduced immune response. 3O-C12-HSL also hinders lymphocyte proliferation [54]. Likewise, ExoU, a phospholipase produced by PA, is reported to be responsible for rapid phagocyte cell death [46]; however, in a more complex model of a three-dimensional epithelial cell and macrophage co-culture, this effect was not observed [79].

In the context of CF, SA and PA often persist as biofilm which protects against antibiotics and host immune cells. Biofilms also hinder proinflammatory immune response [80]. For example PA in biofilm produces rhamnolipids when in contact with neutrophils [81,82]. These rhamnolipids induce neutrophil necrosis [66], which results in the release of proinflammatory compounds that triggers greater neutrophil recruitment, but also of DNA and actin that are used to form more biofilm [83]. In addition, Usher et al. have demonstrated that PA pyocyanin is responsible for decreased cyclic adenosine monophosphate (cAMP) concentration in neutrophils, inducing apoptosis [67]. SA in biofilms also produce toxins, such as Hla and leukicidin AB which promote immune cells death [80,84].

## 4. Modulation of Cytokine Levels

Once bacteria are detected by epithelial and host immune cells, pro-inflammatory cytokine secretion is triggered. PA is known as a proinflammatory cytokine inducer via its QS molecules. For example, Mayer et al. found that 3O-C12-HSL was a strong inducer of IL-6, especially in lung epithelial CF line cells (IB3-1 and CuFi) [72], confirming the role of QS molecules in triggering inflammatory response in CF patients [73]. In addition, 3O-C12-HSL is able to stimulate IL-8 production by epithelial airway cells and fibroblasts [74]. However, the effect of 3O-C12-HSL is complex as it also has an anti-inflammatory effect at elevated concentrations, decreasing IL-6, TNFα, and increasing IL-10 production by LPS-activated macrophages [52]. It also able to inhibit IL-2 produced by PBMC, TNFα produced by monocytes [47,55], and IL-12 produced by LPS-activated dendritic cells [54]. Likewise, Bortolotti et al. found that 3O-C12-HSL induces IL-10 and human leukocyte antigen-G (HLA-G), responsible for human immune response inhibition [53]. Other QS molecules, such as 4-hydroxy-2-heptilquinoline (HHQ) and 2-heptyl-3,4dihydroxyquinoline (pseudomonas quinolone signal, PQS), were found to suppress the innate immune responses in the mouse monocyte/macrophage cell line and cells in bronchoalveolar lavage via IL-6 and TNFα inhibition through the NFκB pathway [49], as well as IL-12 inhibition [54].

Thus, PA QS molecules seem to have two opposite effects on host cells: it triggers inflammatory response at low concentration, but it inhibits host immune response at high concentrations. As QS molecules were found at high concentrations in CF sputum, the anti-inflammatory effect of QS molecules is expected to play a role in host evasion and promotion of chronic infections [5]. Bedi et al. showed how PA is able to modulate host immune response by accumulation of QS molecules [85,86,87].

In addition to QS molecules, PA produces many proteins responsible for modulation of the host immunity response, such as the protease LasB that is responsible for an indirect anti-inflammatory effect by degrading pro-inflammatory cytokines such as IL-6 and IL-8 and neutrophil secreted products [50,57]. In addition to pro-inflammatory cytokine destruction, LasB was found to cleave surfactant protein A (SP-A) and D (SP-D), particularly in CF patients were SP-A and SP-B lung levels are lower [62,64]. As SP-A plays a role in opsonization and phagocytosis of numerous pathogens [63], its degradation could promote pathogens persistence inside CF lungs. As does 3O-C12-HSL, LasB may has both pro and anti-inflammatory effects. Indeed, Sun et al. have found that LasB triggers pro-IL-Iβ maturation, leading to increased IL-Iβ levels and inflammation [71]. PA also modulates host immune response via the secretion of outer membrane vesicles (OMVs) containing short interfering RNA, which led to reduction IL-8 secretion by primary human epithelial airway cells [51]. In addition, PA colonization in airways leads to PD-L1 overexpression on circulating monocytes that exhibit impaired inflammation response initiation and antigen presentation, termed endotoxin tolerance [70].

As PA, SA is also able to modulate host immunity. Chekabab et al. reported that SA reduces IL-8 production triggered by PA on immortalized airway epithelial cells (BEAS-2B) and on immortalized bronchial cells homozygous for the ΔF508 CFTR mutation CFBE41o-[16]. As IL-8 possess a predominant role in inflammation and leucocytes chemotaxis, its inhibition may be a key factor leading to chronic infections, notably in CF patients where IL-8 seems to be able to attract more leucocytes than in healthy patients [88]. IL-8 inhibition may be related to SA β-hemolysin [17] and Sae R/S two-component system (TCS), that are able to inhibit IL-8 production [18]. This reduction of IL-8 production is responsible for reduced neutrophil survival, decreased transmigration, and delayed bacterial clearance, favoring chronic infections [17,18]. SA also produces AdsA responsible for adenosine production [89], which is detected by four G-protein-coupled membrane receptors (A1, A2A, A2B, and A3), triggering anti-inflammatory signaling cascades leading to inhibition of cytokines production (Il-1a and IL-10) [15].

## 5. Modulation of Itaconate Immune Response

During airway infection, SA and PA induce an immunometabolic reprogramming of macrophages that results in an airway environment containing abundant immune signaling metabolites. Basically, toll-like receptor (TLR) triggering by bacteria leads to a metabolic switch from oxidative phosphorylation to glycolysis in macrophages [90]. Glycolysis, allowing more energy production, and promotes succinate, reactive oxygen species ROS, and itaconate release in the respiratory airways [91,92]. Succinate oxidation stabilizes the hypoxia-inducible transcription factor-1α (HIF-1α) that enhances IL-1β synthesis and pro-inflammatory response [90]. To limit tissue damage due to inflammation, itaconate is synthetized by myeloid cells. It suppresses succinate oxidation, leading to succinate accumulation in extracellular medium [93] (Figure 1).

Itaconate has been shown to accumulate in the lungs of CF patients during both SA or PA mono- or co-infection, in function of the duration of infection/colonization [39]. It is highly expressed by myeloid cells after infection with PA [94]. Itaconate inhibits SA and PA glycolysis, leading to metabolic adaptation with increased extracellular polysaccharide (EPS) synthesis that is responsible for increased biofilm production [39,40] but which also triggers itaconate synthesis. In addition, PA isolates, after adaptation, are able to use itaconate as an energy source [39]. This use may contribute to limiting the bactericidal activity of itaconate against SA [40,95,96]. Moreover, succinate accumulation (due to presence of itaconate) induces metabolic stress in PA, increasing growth and biofilm production, both responsible for better airway colonization [97]. Ultimately, inhibition of succinate oxidation by itaconate will reduce IL-1β synthesis. In addition, itaconate decreases neutrophil degranulation and reduces inflammation by inhibiting neutrophil glycolysis through modifications of ALDOA, GAPDH, and LDHA [92,98,99,100] (Figure 1).

Thus, both SA and PA counter host immune response by hijacking itaconate metabolism, leading to bacterial host-adaptation, modifying bacterial metabolism, promoting biofilm formation, and limiting inflammation.

## 6. Modulation of Nutritional Immunity

Nutritional innate immunity is defined as the sequestration of essential metal ions by the host to prevent their capture by pathogens such as bacteria; without such ions, bacterial metabolism is blocked and thus the host prevents bacterial proliferation. Lactoferrin, lipocalin-2, haptoglobin, hemopexin, and calprotectin are the proteins responsible for nutritional immunity [101]. SA possesses two superoxide dismutase (SOD), SodA and SodM, essential for protection against oxidative stress, which are, respectively, manganese and manganese or iron-dependent [25,26,27]. During inflammation, host calprotectin sequesters manganese, impairing SodA activity; SodM escapes this inhibition by using iron, highlighting its importance in SA. To fight against zinc starvation, SA produces staphylopine, a metallophore enabling successful competition for zinc [41]; PA produces pseudopalin, a staphylopine analogue [102]. To counter iron starvation, SA and PA are able to modify their iron intake system; PA is able to switch from Fe^3+^ intake to heme intake in chronic lung infections [68,69], whereas SA prefers heme as the iron source during infection [42]. When co-cultivated in iron depleted media, PA secretes alkylhydroxyquinolones to kill SA to steal its iron [103]. However, in the presence of calprotectin, which is present at high concentrations in CF sputum and inhibits both SA and PA iron uptake [101], the anti-staphylococcal effect of PA is reduced [104]. In particular, Vermilyea et al. found that the activity of LasA and of LasB was inhibited in the presence of calprotectin [105]. Similar results were observed using mice, and in CF lung explants [104]. Thus, calprotectin tends to temper nutritive competition between SA and PA. To fight against calprotectin nutritional immunity, SA possesses two TCS: ArlRS, responsible for global virulence regulation [43], and SaeRS that is activated by calprotectin [106]. SaeRS is responsible for the regulation of more than 20 staphylococcal virulence factors, including anti-neutrophils factors [13,24]. Taken together this suggest that, in the presence of calprotectin, PA is able to modulate its anti-staphylococcal effect while SA adapts its metabolism to favor its growth in a nutriment depleted environment, thus favoring SA/PA coinfections; this may also explain why SA and PA colocalize with regions where calprotectin is highly expressed [104].

## 7. Evasion to Neutrophil Activities

The inflammatory response in CF airways is dominated by a massive influx of neutrophils. Neutrophil recruitment aims to regulate infection by (i) massive release of antimicrobial enzymes from granules such as myeloperoxidase, neutrophil elastase and lactoferrin, (ii) neutrophil extracellular traps (NETs), and (iii) phagocytosis. Both SA and PA are able to counteract neutrophil activities (Figure 2).

SA and PA are able to escape neutrophil recognition by biofilm (MucA) and pseudo capsule formation (CoA, vWBO). They are able to inhibit ROS production (Siglec 9 binding, AdsA) and to limit ROS effect (AhoC, catalase, SodA, SodM, carotenoid). SA and PA also have the ability to inhibit neutrophil degranulation (Siglec 9 binding, AdsA, CHOP) and to block neutrophils products activity (eap, aureolysin, EapH1, EapH2, staphylokinase). Finally, SA and PA are able to inhibit NETs formation (Siglec 9 binding, loss of flagellum/motility, LasR deficiency, NucA, AdsA). SA and PA may impair neutrophil response by inducing neutrophils death (γ hemolysin, PVL, rhamnolipids). Moreover, SA is able to inhibit chemotaxis and complement activation (Eap, CHIPS, SCIN, ecb, efb).

### 7.1. Neutrophil Recognition Evasion

To evade cellular immune responses, SA and PA are able to hide from neutrophils by producing biofilm [20], and PA mucA mutants, characterized by a high alginate production, decreased neutrophils attraction, and complement activation, are frequently isolated in CF lungs [56]. In addition to biofilm production, SA is also able to produce a pseudo capsule using its two coagulases, CoA and vWBP, impairing neutrophil access [19], and it is able to inhibit chemotaxis and complement activation with several virulence factors [extracellular adherence protein (Eap), staphylococcal complement inhibitor (SCIN), chemotaxis inhibitory protein of staphylococci (CHIPS), extracellular complement binding protein (Ecb), and extracellular fibrinogen binding protein (Efb)] leading to decreased neutrophil activity [10,11,12,13,14,20,37].

### 7.2. Degranulation Evasion

SA AdsA inhibits neutrophil degranulation and oxidative burst [15,21]; it also secretes three neutrophil serine protease inhibitors, namely Eap, EapH1, and EapH2, which are able to inhibit neutrophil proteases such as proteinase 3 and cathepsin G [22]. As described above, SA also secretes two super oxide dismutases, SodA and SodM, that protect itself from neutrophil-induced oxidative stress [25,26,27]. Treffon et al. found that SodM was overexpressed during CF chronic infections [26]. In addition, SA also produces a catalase, KatG, and an alkylhydroperoxide reductase, AhpC, protecting SA against hydrogen peroxide [29]. Another SA product, the carotenoid pigment, which harbors an antioxidant activity, was shown to protect SA against neutrophil killing [28]. SA is also able to protect itself against alpha-defensin via staphylokinase [24] and against the antimicrobial peptide cathelicidin LL-37 via aureolysin [23].

PA protects itself from antimicrobial peptide production via the stimulation of the UPR regulation pathway and production of CHOP as described above, although this mechanism also leads to enhanced ROS production [85]. However, PA inhibits elastase and ROS production via binding to siglec-9 at the surface of neutrophils [58]. Siglec are transmembrane proteins that bind sialylated carbohydrates on targeted cells to regulate binding, cell proliferation, cell signaling, endocytosis, and natural killer-mediated cell lysis.

### 7.3. NETosis Evasion

SA and PA have also developed mechanisms to counter NETosis, which is the release of NETs composed of an extracellular DNA backbone associated with antimicrobial peptides (notably calprotectin), histones and proteases by neutrophils to capture and kill bacteria [107,108]. Gray et al. found that NETosis was enhanced in CF patients, making it a critical host defense mechanism [109]. In addition to intrinsic enhanced NETosis, PA also moderately induces formation of NETs via the protease LasA [75], and more importantly by its flagellum and its motility [59]. SA also induces activation of NETs [31].

Both SA and PA have developed mechanism to escape NETosis. For instance, chronic PA strains are often LasR-deficient leading to the loss of LasA and LasB, and frequently lose its flagellum leading to reduced activation of NETs [59,60]. Another PA mechanism of NET inhibition is its binding to siglec-9 on the surface of neutrophils [58]. SA produces nuclease that is able to degrade NETs [31,32]. High nuclease-producing SA strains are selected by environments with high inflammation as seen in CF [110], and such bacteria have been reported to induce a delay in bacterial clearance and enhance mortality in an in vivo murin model of SA respiratory tract infection [32]. In addition, SA AdsA is also able to degrade NETs, and its activity is potentiated by staphylococcal nuclease [30].

## 8. Phagocytosis Evasion

Phagocytosis is a common strategy used by neutrophils and macrophages to eradicate bacteria. SA and PA have developed strategies to escape phagocytosis. SA AdsA is responsible for adenosine production, which downregulates phagocytosis by alveolar macrophages through A2aR/A2bR (in particular A2aR)—PKA pathways and modulation of p38 phosphorylation [33,34]. SA also produces SpA, which binds to the Fcγ domain of immunoglobulins, leading to phagocytosis inhibition [15,35]. Most SpA is cell wall-anchored, but a fraction of SpA is also secreted; Armbuster et al. demonstrated that this secreted SpA could also protect PA from phagocytosis [111]. Another SA protein, staphylococcal binder of immunoglobulin (Sbi), is also able to bind the Fcγ domain of immunoglobulins, then consumes C3 and inhibits phagocytosis [36].

In order to inhibit phagocytosis, PA is able to inject toxins inside host cells via its T3SS. Among these toxins, ExoT and ExoS, two GTPase-activating proteins, have been found to inhibit macrophage phagocytic capabilities by interfering with cytoskeletal rearrangement [46]. In addition to toxins, PA LasB is able to disarm host-protease-activated receptors 2 (PAR2), a lung inflammation regulator, inducing a reduced bacterial clearance probably through phagocytosis inhibition [65]. PA phagocytic evasion is also promoted by loss of motility, a phenotype frequently observed during chronic infections [61,112].

## 9. How Host Immune Response Modulation Promotes SA-PA Coinfection

As discussed above, both SA and PA possess many factors capable of modulating and inhibiting the host immune response (Table 1 and Table 2) and thus favor their persistence and the establishment of chronic infections. By different and complementary mechanisms, both species (i) inhibit the proliferation of immune cells or induce their apoptosis, (ii) inhibit the production of pro-inflammatory cytokines, and (iii) counteract the immune response by developing resistance strategies (i.e., increase in biofilm production, metabolic modification to resist oxidative stress or nutritional deficiencies, hijacking of neutrophil actions, among others). The consequent inhibition of the immune response is not specific to a species, and benefits not only the species that implements them but also the others present, potentially promoting coexistence of both pathogens in CF airways. For example, SA has been reported to demonstrate anti-inflammatory activity (reduction of IL-8 production) when co-cultivated with PA [16], suggesting cooperation between SA and PA to promote chronic infection.

Furthermore, some mechanisms developed only by one species still benefit both species. Thus, PA-induced inhibition of mucociliary clearance logically promotes SA colonization. Conversely, Wieneke et al. have found that the high nuclease producer SA were associated with PA coinfection, suggesting that nuclease, via its host immunity modulation activity is critical to promote SA-PA coinfection in CF airways [110].

The relationship between different microorganisms and the immune response may be more complex than it appears. It has been described that PA is able to highjack host immune response to outcompete other microorganisms and dominate the microenvironment. PA induces the production of sPLA2-IIA, an antimicrobial peptide, in CF epithelial cells via a T3SS-dependent process; PA inject ExoS into host cells via its T3SS, leading to Krüppel-like factor 2 (KLF2) activation, and then sPLA2-IIA production. sPLA2-IIA is found at high concentration in sputum from CF patients, sufficient to kill SA but not to kill PA [33]. Of note sPLA2-IIA reaches its highest concentration in early adulthood, when the dogma of SA colonization to PA colonization switch stands. However, SA has mechanism to resist sPLA2-IIA thanks to adenosine production that inhibits that of sPLA2-IIA [15,34], and in vivo data from a pulmonary infection of guinea pig model indicate that adenosine production decreases SA clearance from airways [34]. Therefore, this mechanism, initially expected to be responsible for SA eradication in CF airways, may finally promote SA–PA coinfections, at the expense of other microorganisms.

These different examples show how modulation of the immune response would be an actor in the establishment of coinfections, especially in CF lungs where about 40% of the patients are coinfected by these bacteria. Moreover, studies tend to show that SA/PA coinfections in CF patients lead to a less severe pulmonary condition than PA alone, suggesting that SA/PA coinfection may reduce host immune response [113,114]. However, this synergy and the idea that modulation of the immune response by one species benefits other species remains to be demonstrated by in vitro approaches.

## 10. Future Directions

Most of the mechanisms of immune response modulation described in this review have been explored in vitro, mainly using laboratory strains. Only a few studies have explored the behavior and impact of CF patient strains [16,26,27,31,33,40,50,56,62,69,85,97]. However, it is known that clinical strains present a significant genomic and phenotypic diversity. It would, therefore, be interesting to complete these studies using strains from CF patients. Furthermore, it has been clearly described that strains of PA [115,116,117,118,119] and SA [26,31,120,121,122,123,124] evolve during chronic infection to adapt to the lung environment and persist over time. In PA, this adaptive evolution leads to the establishment of high antibiotic resistance, increased biofilm forming capacity, slowed metabolism, and decreased virulence. In SA, this evolution is accompanied by profound metabolic changes and frequent acquisition of the small colony variant (SCV) phenotype [125], as well as improved resistance to ROS and NETs [26,31]. Thus, these evolutions may directly affect the mechanisms of immune response modulation described in this review. It would therefore be relevant to analyze longitudinally the impact of SA and PA on the host response from CF clinical strains isolated at different time points.

Finally, in this review we only looked at SA and PA interactions with the host immune system. As SA and PA are part of the CF pulmonary microbiome, we could expect that there are more interactions between the microbiome and the host immune system, which remain to be assessed.

## 11. Conclusions

SA and PA are able to bypass the physical integrity of airway epithelia, evade recognition, and then modulate host immune cell proliferation. They also modulate the immune response by regulating cytokine production and by counteracting the activity of neutrophils and other immune cells. Inhibition of the immune response benefits not only the species that implements them but also other species present, and therefore, it can promote the establishment of coinfections in CF lungs.

## Figures and Tables

**Figure 1 ijms-24-06609-f001:**
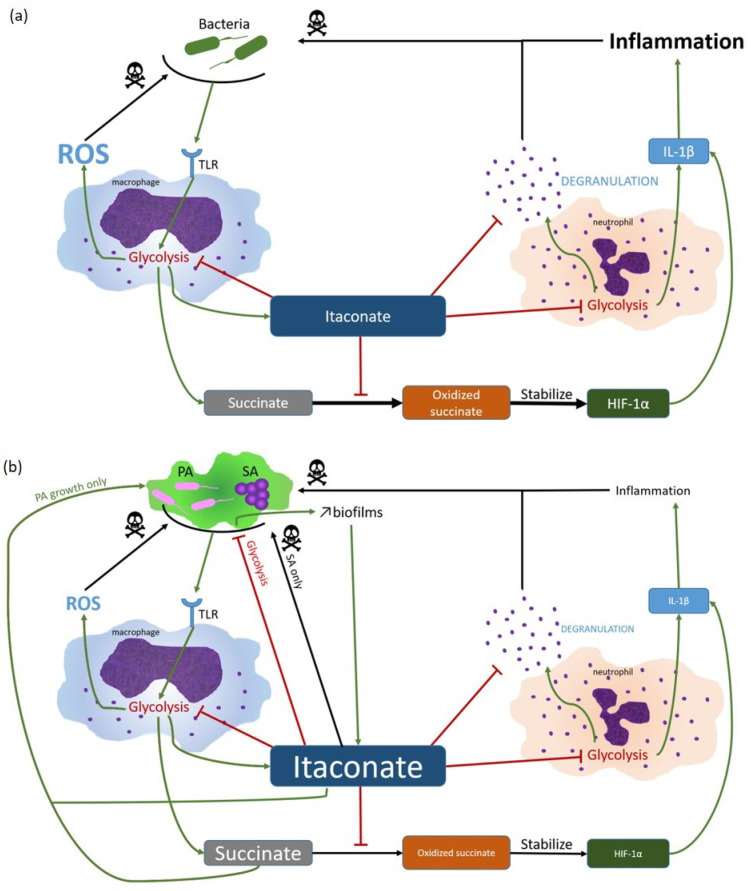
Itaconate hijacking by SA and PA. (**a**) In normal circumstances, SA and PA are detected by TLR, leading to macrophage activation and inflammation induction via succinate oxidation. In order to modulate pro-inflammatory responses, itaconate is synthetized. Itaconate inhibits glycolysis, succinate oxidation and neutrophils degranulation, drastically reducing inflammation and protecting host cells. (**b**) During chronic infections, SA and PA are able to adapt and hijack host response. Despite bacterial glycolysis inhibition, itaconate induces PA growth by succinate accumulation which is used as energy source. Secondarily, itaconate leads to more EPS synthesis, enabling more biofilm formation, which in turn induces itaconate production, promoting SA PA persistency.

**Figure 2 ijms-24-06609-f002:**
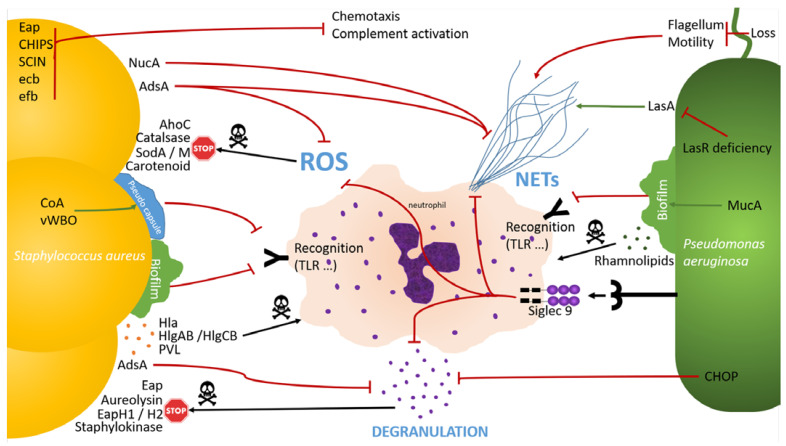
SA and PA anti-neutrophils effects, phagocytosis excluded.

**Table 1 ijms-24-06609-t001:** *Staphylococcus aureus* impacts on host immunity.

Effect	Effector	Bibliography
**Anti-inflammatory effect**		
Epithelium lysis	Hla	Seilie et al. [9]
Chemotaxis and complement activation inhibition	Eap	Chavakis et al. [10]
	CHIPS	de Haas et al. [11]; Rooijakkers et al. [12]
Complement activation inhibition	SCIN	Rooijakkers et al. [13]
	ecb	Jongerius et al. [14]
	efb	Jongerius et al. [14]
IL-1a inhibition	AdsA	Thammavongsa et al. [15]
IL-8 inhibition	Unknown	Chekabab et al. [16]
	β haemolysin	Tajima et al. [17]
	Sae R/S	Zurek et al. [18]
IL-10 inhibition	AdsA	Thammavongsa et al. [15]
Neutrophil recognition evasion	CoA	Guggenberger et al. [19]
	vWBP	Guggenberger et al. [19]
	Biofilm	Parker et al. [20]
Neutrophil degranulation inhibition	AdsA	Thammavongsa et al. [15,21]
Neutrophil proteases inhibition	Eap, EapH1 and EapH2	Stapels et al. [22]
Neutrophil products degradation	Aureolysin	Sieprawska-Lupa et al. [23]
Neutrophil products protection	Staphylokinase	Jin et al. [24]
Oxidative burst inhibition	AdsA	Thammavongsa et al. [18,19,20,21,22,23,24,25,26,27,28,29,30,31,32,33,34,35,36,37,38,39,40,41,42,43,44,45,46,47,48,49,50,51,52,53,54,55,56,57,58,59,60,61,62,63,64,65,66,67,68,69,70,71,72,73,74,75,76,77,78,79,80,81,82,83,84,85,86,87,88,89,90,91]
ROS inhibition	SodA and SodM	Garcia et al. [25]; Treffon et al. [26,27]
	Carotenoid	Liu et al. [28]
	KatG	Cosgrove et al. [29]
NETs degradation	AdsA	Thammavongsa et al. [30]
	Nuclease	Herzog et al. [31]; Berends et al. [32]
Phagocytosis inhibition	AdsA	Pernet et al. [33,34]
	SpA	Falugi et al. [35]
	Sbi	Kim et al. [36]
Phagocytic cells killing	Hla HlgAB HlgCB	Seilie et al. [9] Foster et al. [37]; Seilie et al. [9] Seilie et al. [9]
	PVL	Foster et al. [37]
	AdsA	Winstel et al. [38]
T-cell activation inhibition through adenosine accumulation	AdsA	Thammavongsa et al. [15]
B-Lymphocytes apoptosis	SpA	Kim et al. [36]
sPLA2-IIA inhibition	AdsA	Pernet et al. [33,34]
Itaconate induction	Biofilm	Riquelme et al. [39]; Tomlinson et al. [40]
Nutritional immunity resistance	Staphylopine	Grim et al. [41]
	Heme intake	Skaar et al. [42]
Increased growth in presence of calprotectin	ArlRS	Radin et al. [43]
Activated in presence of calprotectin	Sae R/S	Rooijakkers et al. [13]; Jin et al. [24]
**Pro inflammatory effect**		
NETs formation	Unknown	Herzog et al. [31]

Abbreviations: IL, Interleukin; ROS, reactive oxygen species; NETs, neutrophil extracellular traps; sPLA2-IIA, group II secretory phospholipase A2.

**Table 2 ijms-24-06609-t002:** *Pseudomonas aeruginosa* impacts on host immunity.

Effect	Effector/Pathway	Bibliography
**Anti-inflammatory effect**		
Mucus thickening and mucociliary clearance impairing	Cif	Stanton et al. [44]
Epithelium lysis	3O-C12-HSL	Schwarzer et al. [45]
	Exo U	Hauser et al. [46]
IL-2 inhibition	3O-C12-HSL	Hooi et al. [47]
IL-6 inhibition	3O-C12-HSL	Li et al. [48]
	HHQ	Kim et al. [49]
	PQS	Kim et al. [49]
IL-6 destruction	LasB	LaFayette et al. [50]
IL-8 inhibition	OMVs	Koeppen et al. [51]
	UPR/CHOP/PPARγ pathway	Bedi et al. [39,40,41]
IL-8 destruction	LasB	LaFayette et al. [50]
IL-10 induction	3O-C12-HSL	Glucksam-Galnoy et al. [52], Bortolotti et al. [53]
IL-12 inhibition	PQS	Skindersoe et al. [54]
	3O-C12-HSL	Skindersoe et al. [54], Telford et al. [55]
TNFα inhibition	HHQ	Kim et al. [49]
	PQS	Kim et al. [49]
	3O-C12-HSL	Hooi et al. [47]
Neutrophil recognition evasion	Biofilm	Pedersen et al. [56]; Parker et al. [20]
Neutrophil products degradation	LasB	Skopelja et al. [57]
Neutrophil protease inhibition	siglec-9 binding	Khatua et al. [58]
ROS inhibition	siglec-9 binding	Khatua et al. [58]
	UPR/CHOP/PPARγ pathway	Bedi et al. [39,40,41]
NETs inhibition	siglec-9 binding	Khatua et al. [58]
	LasR deficiency	Floyd et al. [59]; Skopelja-Gardner et al. [60]
Phagocytosis inhibition	Loss of motility	Lovewell et al. [61]
	Exo T	Hauser et al. [46]
	Exo S	Hauser et al. [46]
	LasB	Mariencheck et al. [62]; Kuang et al. [63]; Alcorn et al. [64]; Moraes et al. [65]
PBMC, mast cells and lymphocyte proliferation inhibition	3O-C12-HSL	Hooi et al. [47]; Li et al. [48]; Skindersoe et al. [54]
	PQS	Hooi et al. [47]; Li et al. [48]
Phagocytic cells killing	Exo U	Hauser, [46]
	Rhamnolipids	Jensen et al. [66]
	Pyocyanin	Usher et al. [67]
Itaconate induction	Biofilm	Riquelme et al. [39]
Nutritional immunity resistance	Heme intake	Reinhart et al. [68]; Nguyen et al. [69]
QS molecule accumulation	Paraoxoanse-2	Bedi et al. [39,40,41]
Endotoxin tolerance	Unknown	Avendaño-Ortiz et al. [70]
**Pro inflammatory effect**		
IL-1B induction	LasB	Sun et al. [71]
IL-6 induction	3O-C12-HSL low concentration	Mayer et al. [72]; Li et al. [48]
IL-8 induction	3O-C12-HSL low concentration	Shiner et al. [73]; Smith et al. [74]
NETs formation	LasA	Gambello et al. [75]
	Motility	Floyd et al. [59]
	Flagellum	Floyd et al. [59]
sPLA2-IIA induction	ExoS	Pernet et al. [33]

Abbreviations: IL, interleukin; TNF, tumor necrosis factor; ROS, reactive oxygen species; NETs, neutrophil extracellular traps; PBMC, peripheral blood mononuclear cell; QS, quorum sensing; sPLA2-IIA, group II secretory phospholipase A2.

## Data Availability

No new data were created or analyzed in this study. Data sharing is not applicable to this article.

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
