# Peer review of "How Staphylococcus aureus and Pseudomonas aeruginosa Hijack the Host Immune Response in the Context of Cystic Fibrosis"

_ijms, 2023, doi:10.3390/ijms24076609_

Round 1
Reviewer 1 Report
This review focusses on the measures employed by both Staphylococcus aureus and Pseudomonas aeruginosa to evade and use the host immune system to survive and to cause chronic infections in cystic fibrosis patients. It is a comprehensive and detailed review which is divided into sections making it accessible and clear. The figures are nicely presented, and the tables provide a useful summary of the various pathways/effectors. I think the review may benefit further from the inclusion of recent publications such as: Bernardy EE, Raghuram V, Goldberg JB. Staphylococcus aureus and Pseudomonas aeruginosa Isolates from the Same Cystic Fibrosis Respiratory Sample Coexist in Coculture. Microbiol Spectr. 2022 Aug 31;10(4):e0097622, and also Price CE, Brown DG, Limoli DH, Phelan VV, O'Toole GA. Exogenous Alginate Protects Staphylococcus aureus from Killing by Pseudomonas aeruginosa. J Bacteriol. 2020 Mar 26;202(8):e00559-19. Also, if it has not been mentioned, it might be useful to add a sentence of two on the role of interactions with the wider CF microbiome. Other than this I only have minor suggested changes.
Line 19: Inhibition of the immune response benefits not only the species that implements them.
Line 33: SA are the most 33 prevalent bacteria
Line 37: “Between 8.6% to 60% of CF patients are co-colonized by the two bacteria”. It might be better to write “It is estimated that between 8.6% to 60% ….”.
Table 1: IL-8 inhibition-presumably the anti-inflammatory factor described in Chekabab et al., 2015 is unknown. It would be better to write this rather than adding a question mark. The same for “NETs formation” (Herzog et al., 2019).
It would be useful to have a definition of “NETs” and other abbreviations in the table footer.
Table 2: The same as above for Endotoxin tolerance in P. aeruginosa. Please change the question and provide further detail or write “unknown”. Consider whether it would be better to add these unknown factors to the footnotes of the tables.
Line 87: In the context of CF, SA and PA often persist
Line 157: “Favouring” may be more appropriate than “favorizing”.
Author Response
We thank the reviewer for the constructive comments. We have tried to take the comments into consideration in order to bring more clarity to the manuscript.
We are aware that this manuscript is rich in information as the interactions of SA and PA with the host are numerous and involve many factors. It was our ambition to make a comprehensive review of current knowledge. In order to make it clear, we have chosen to make two tables that summarize all the bacterial factors and their actions on the host. For the more complex passages, we have also added figures (figures 1 and 2). It is not easy to illustrate in a single figure all the mechanisms described in paragraphs 2 and 3, as they involve many bacterial factors and many host targets. The illustration in the form of a table seems more appropriate to us.In paragraph 3, we have simplified the sentence mentioned by the evaluator for clarity. We have also simplified paragraph 4 to make it more understandable, as suggested.
Changes to the manuscript are shown in blue. We hope that this will meet the expectations of the reviewers
Reviewer 2 Report
Major point: The review by Souche and coworkers is overall interesting and exhaustive. The Authors made a comprehensive review of the literature, highlighting many mechanisms of host immune response evasion by S. aureus and P. aeruginosa in the context of cystic fibrosis. Nevertheless, the huge amont of data reported, the complex mechanisms involved, and the numerous pathways referred to make the review sometimes difficult to follow.
In my opinion the Authors should try to simplify even by sacrificing some information but increasing clarity and incisiveness.
Some specific points follow.
In paragraph 2 and 3 Authors refer to several virulence factors of SA and PA. A Figure summarizing in a schematic manner the multitude of virulence factors and their effects on immune cells would help the readers not fully expert in the field.
Paragraph 3, sentence "however, Crabbé et al. 83 demonstrated on a 3 dimensional epithelial cell and macrophage co-culture model that 84 3O-C12-HSL -mediated apoptosis was inhibited". Please explain better; apoptosis was inhibited as compared to what? How do the different models the authors refer to may impact the results obtained in different studies?
Paragraph 4: Please revise the sentence "This may be related to SA β-hemolysin that is able to 153 inhibit IL-8 production, without cytotoxic activity against endothelial cells [55], but also 154 another virulence factor, Sae R/S two-component system (TCS)"; it is very hard to understand; too many concepts in a sentence.
In my opinion Paragraph 4 is very hard to follow. It would help if the authors could elaborate the information in an integrated and plausible model rather than simply make a list of the numerous and often contradictory findings of the literature.
Author Response
In contrast, we thank the reviewer for the constructive comments. We have tried to take the comments into consideration in order to bring more clarity to the manuscript.
Regarding the two publications that the reviewer suggests to include in the manuscript, it seems to us that they are not related to the subject matter. Although very interesting, they deal with the interactions between SA and PA and how the two species can coexist. However, these two studies do not focus on the interaction of these two pathogens with the host. In this sense, we do not think it is necessary to include them in this manuscript.
On the contrary, we agree with the reviewer that SA and PA are not the only actors in host-pathogen interactions and are elements of a more complex microbiota whose interaction with the host remains to be explored. We have added a short paragraph in this sense in chapter 10: "future directions".
Finally, all suggested minor changes have been included in the new manuscript.
Changes to the manuscript are shown in blue. We hope that this will meet the expectations of the reviewers